# Inhibition of Granule Cell Dispersion and Seizure Development by Astrocyte Elevated Gene-1 in a Mouse Model of Temporal Lobe Epilepsy

**DOI:** 10.3390/biom14030380

**Published:** 2024-03-20

**Authors:** Eunju Leem, Sehwan Kim, Chanchal Sharma, Youngpyo Nam, Tae Yeon Kim, Minsang Shin, Seok-Geun Lee, Jaekwang Kim, Sang Ryong Kim

**Affiliations:** 1BK21 FOUR KNU Creative BioResearch Group, School of Life Science, Kyungpook National University, Daegu 41566, Republic of Korea; leem@kmedihub.re.kr (E.L.); arputa@knu.ac.kr (S.K.); chanchalmrt@gmail.com (C.S.); taetaey@hanmail.net (T.Y.K.); 2Dementia Research Group, Korea Brain Research Institute, Daegu 41062, Republic of Korea; 3Efficacy Evaluation Department, New Drug Development Center, Daegu-Gyeongbuk Medical Innovation Foundation (K-MEDI Hub), Daegu 41061, Republic of Korea; 4Brain Science and Engineering Institute, Kyungpook National University, Daegu 41944, Republic of Korea; blackpyo2@knu.ac.kr; 5Department of Microbiology, School of Medicine, Kyungpook National University, Daegu 41944, Republic of Korea; shinms@knu.ac.kr; 6Department of Biomedical Science & Technology and BioNanocomposite Research Center, Kyung Hee University, Seoul 02447, Republic of Korea; seokgeun@khu.ac.kr

**Keywords:** adeno-associated virus 1, adeno-associated viral vector, astrocyte elevated gene-1, gene therapy, granule cell dispersion, hippocampus, kainic acid, mammalian target of rapamycin complex 1, seizure, temporal lobe epilepsy

## Abstract

Although granule cell dispersion (GCD) in the hippocampus is known to be an important feature associated with epileptic seizures in temporal lobe epilepsy (TLE), the endogenous molecules that regulate GCD are largely unknown. In the present study, we have examined whether there is any change in AEG-1 expression in the hippocampus of a kainic acid (KA)-induced mouse model of TLE. In addition, we have investigated whether the modulation of astrocyte elevated gene-1 (*AEG-1*) expression in the dentate gyrus (DG) by intracranial injection of adeno-associated virus 1 (AAV1) influences pathological phenotypes such as GCD formation and seizure susceptibility in a KA-treated mouse. We have identified that the protein expression of AEG-1 is upregulated in the DG of a KA-induced mouse model of TLE. We further demonstrated that *AEG-1* upregulation by AAV1 delivery in the DG-induced anticonvulsant activities such as the delay of seizure onset and inhibition of spontaneous recurrent seizures (SRS) through GCD suppression in the mouse model of TLE, while the inhibition of *AEG-1* expression increased susceptibility to seizures. The present observations suggest that *AEG-1* is a potent regulator of GCD formation and seizure development associated with TLE, and the significant induction of *AEG-1* in the DG may have therapeutic potential against epilepsy.

## 1. Introduction

Epilepsy is a neurological disorder that affects more than 65 million people worldwide [1]. According to recent meta-analysis studies, the prevalence of active epilepsy is around 6.4 per 1000 persons, and the annual incidence is around 67.8 per 100.000 [2]. Epilepsy is characterized by spontaneous recurrent seizures (SRS) and associated with cognitive, psychological, and social problems [3,4]. An epileptic seizure is a transient behavioral change or symptom caused by abnormal excessive or asynchronous neuronal activity in the brain [5]. Among various epilepsy types, temporal lobe epilepsy (TLE) is the most common form of focal epilepsy. TLE is characterized by hippocampal sclerosis and SRS. The hippocampal sclerosis of TLE is frequently associated with neuronal loss in the CA1, CA3, and hilar regions of the dentate gyrus (DG) and granule cell dispersion (GCD), which is the abnormal transformation of the granule cell layer induced by epileptic insults, in the DG region [5,6,7]. Although there has been a significant advance in our understanding of epilepsy, the molecular mechanisms underlying epilepsy remain elusive [4]. Thus far, dozens of anti-seizure drugs (ASDs) have been approved for prescription in clinics. However, there is an urgent need to identify additional molecular targets for epilepsy therapy, since in one-third of epilepsy cases, medicines failed to control seizures [3].

Astrocyte elevated gene-1 (*AEG-1*), also known as metadherin (*MTDH*), is a well-established oncogene that plays critical roles in multiple tumors [8,9]. AEG-1 is highly expressed in the brain, especially in neurons and neuronal precursor cells [10]. AEG-1 participates in alternative splicing processes, inhibiting factors that disrupt the absorption of free fatty acids, which ultimately facilitates fat uptake [11,12]. Additionally, AEG-1 possesses an LxxLL motif, which modulates gene expression by interacting with transcriptional factors and cofactors under normal physiological conditions [9,13,14]. In addition to cancer, a growing body of evidence has suggested that AEG-1 may play crucial roles in the pathogenesis of various neurological disorders. *AEG-1* was originally identified as a neuropathology-associated gene in human fetal astrocytes that was induced by human immunodeficiency virus-1 and tumor necrosis factor [15,16,17]. A recent genome-wide study showed that *AEG-1* expression is associated with migraine [18]. Our group recently demonstrated that AEG-1 is reduced in a mouse model of Parkinson’s disease [19]. AEG-1 upregulation exerts a protective effect against 6-hydroxydopamine (6-OHDA) in the dopaminergic neurons in vivo [19]. Moreover, AEG-1 is also reduced in motor neurons of an amyotrophic lateral sclerosis (ALS) mouse model [20]. Silencing *AEG-1* induces apoptotic cell death by inhibiting phosphatidylinositol-3-kinase (PI3K)/protein kinase B (AKT) signaling in mouse motor neuronal NSC34 cells [20]. However, it has never been investigated whether AEG-1 is involved in the pathogenesis of epilepsy. In the present study, we first examined whether AEG-1 expression is regulated by kainic acid (KA) in the hippocampus of the mouse brain. We next investigated whether the modulation of AEG-1 expression in the DG influences pathological phenotypes such as GCD and seizure susceptibility in a KA-induced mouse model of TLE and further examined the potential mechanism associated with AEG-1-mediated effects against TLE.

## 2. Materials and Methods

### 2.1. Animals and Pharmacological Administration

Male C57BL/6 mice (postnatal 8 weeks, 23 ± 2 g) were purchased from Daehan Biolink (Eumseong, Daegu, Republic of Korea). The mice were raised in an isolated environment with standard chow and water provided ad libitum. Before the stereotaxic injection, mice were anesthetized with 115 mg/kg of ketamine, chemically known as 2-(2-chlorophenyl)-2-(methylamino)cyclohexanone hydrochloride (Yuhan, Daegu, Republic of Korea), and 23 mg/kg of rompun, composed of xylazine hydrochloride and methyl hydroxybenzoate (Bayer Korea Ltd., Daegu, Republic of Korea) [21]. Rompun, a non-narcotic compound, is used in veterinary medicine as a sedative, analgesic, muscle relaxant, and enhancement of ketamine. KA (Cat. No. K0250; Sigma-Aldrich, St. Louis, MO, USA) was dissolved in phosphate-buffered saline (PBS) to a concentration of 1 µg/µL. Each mouse was administrated 2 μL of KA in the right hippocampal CA1 layer by stereotaxic injection [anterior-posterior (AP): −2.0 mm; medial-lateral (ML): −1.2 mm; dorsal-ventral (DV): −1.5 mm, relative to the bregma] using a 30-gauge Hamilton syringe needle at a rate of 0.5 µL/min [7,22,23]. After injection, the needle was left in place for 5 min and retracted slowly to prevent reflux. For rapamycin administration, each mouse was intraperitoneally injected with 10 mg/kg of rapamycin (LC Laboratories, Woburn, MA, USA) in a solution containing 4% ethanol, 5% Tween 80, and 5% polyethylene glycol 400 or vehicle once per day from one day before KA injection to 6 days post-lesion, as described [24,25,26,27].

### 2.2. Intracranial Injection of AAVs

The viral vectors were manufactured as described previously with some modifications [19,28,29]. Briefly, AEG-1 with a hemagglutinin (HA)-tag at the 3′-end and green fluorescence protein (GFP) was cloned into the pBL plasmid and packaged into adeno-associated virus serotype 1 (AAV1) capsid to generate AAV1-AEG-1-HA and AAV1-GFP viruses, respectively. AAV1-AEG-1 shRNA (shAEG-1)-EGFP and AAV1-scrambled control (shCON)-EGFP viruses were purchased from Goma Biotech (Goma Biotechnology, Seoul, Republic of Korea). The sequences of shRNA targeting *AEG-1* and scrambled control are as follows: 5′-CCGGAGCCATCTATTACCTTATCAACTCGAGTTGATAAGGTAATAGATGGCTTTTTTG-3′ for shAEG-1 and 5′-GAATTCGCTTCCTTCTTGCGTAAGTTTTCAAGAGAAACTTACGCAAGAAGGAAGC-3′ for shCON. The delivery of AAVs was performed as previously described with some modifications [21,26,28,29]. Briefly, 8-week-old mice were anesthetized with 115 mg/kg of ketamine and 23 mg/kg of xylazine and placed in a stereotaxic frame (Kopf Instruments, Tujunga, CA, USA) with a mouse adapter. Each mouse received a unilateral injection of AAV into the right DG (AP: −2.0 mm, ML: −1.4 mm, DV: −1.8 mm, relative to the bregma) using a 30-gauge Hamilton syringe attached to an automated microinjector [23]. The virus suspension in a volume of 2 µL of 9.4 × 10^12^ viral genomes/mL was injected at a rate of 0.1 µL/min over 20 min. After injection, the needle was left in place for 5 min and retracted slowly to prevent reflux.

### 2.3. Monitoring of KA-Induced Seizures

The mice were recorded for 4 h following KA treatment, and their behaviors were observed and noted to evaluate the latency of seizure onset, total duration, and severity, as previously described with some modification [7,22]. The severity of seizures was classified into five stages according to a modified scale, as follows: stage 1, characterized by facial movements; stage 2, characterized by head nodding and myoclonic twitching; stage 3, characterized by forelimb clonus with lordotic posture; stage 4, characterized by forelimb clonus with reared posture; and stage 5, characterized by tonic–clonic seizures without postural control. In this study, the mice exhibiting a severity of stage 3 or higher were considered positive for seizure onset [7,22]. In addition, three weeks after KA treatment, the mice were placed in uncovered cages enclosed with white acrylic walls. The mice were monitored by video recording for two weeks (9 h/day, 6 days/week) to assess SRS [7,22].

### 2.4. Tissue Preparation

The mice were anesthetized using a ketamine/xylazine mixture and transcardially perfused with 4% paraformaldehyde in 100 mM PBS (4% PFA). The extracted brains were post-fixed in 4% PFA at 4 ℃ overnight and then cryoprotected in 30% sucrose containing 4% PFA until processed. The fully fixed brains were frozen and cut into 30-µm-thick coronal sections using an HM525 NX cryostat (Thermo Fisher Scientific, Waltham, MA, USA), and the sections were gathered from −1.07 to −2.53 mm posterior to bregma according to the mouse brain atlas, as previously described [7,22,24].

### 2.5. Nissl Staining and Assessment of GCD

For Nissl staining, the brain sections (*n* = 5 for each mouse) were mounted on gelatin-coated slides and stained with 0.5% cresyl violet (Sigma-Aldrich). The presence of GCD was assessed by measuring the abnormal distribution of granule cells in the granule cell layer (GCL). In brief, the width of GCL in the DG of each mouse was determined by measuring the distances from the hilar border to the outer border of the upper blade of GCL in 4 alternate sections positioned at −1.67, −1.79, −2.15, and −2.27 mm posterior to the bregma. After section preparation, the first- and third quarter portions of the upper blade of the DG were measured to obtain an average width of GCL in both ipsilateral and contralateral hemispheres. The GCD level was assessed by comparing the GCL width between the experimental groups at 7 days post-lesion, as described previously [7,22,24].

### 2.6. Immunostaining

The primary antibodies used are as follows: mouse anti-neuronal nuclei (NeuN; Millipore, Burlington, MA, USA), rabbit anti-AEG-1 (Invitrogen, Camarillo, CA, USA), and rabbit anti-p-4E-BP1 (Cell Signaling, Beverly, MA, USA). (i) Mouse anti-NeuN (1:500) and rabbit anti-AEG-1 (1:500) for immunohistochemistry, and (ii) mouse anti-NeuN (1:200), rabbit anti-AEG-1 (1:200), and rabbit anti-p-4E-BP1 (1:1000) for immunofluorescence. The secondary antibodies used are as follows: biotin-conjugated anti-mouse IgG (1:400; Kirkegaard and Perry Laboratories Inc., Gaithersburg, MD, USA), biotin-conjugated anti-rabbit IgG (1:400; Vector Laboratories, Burlingame, CA, USA), Texas Red-conjugated anti-mouse IgG (1:400; Vector Laboratories, Burlingame, CA, USA), and FITC-conjugated anti-rabbit IgG(1:400; Jackson ImmunoResearch: Laboratories, Bar Harbor, ME, USA). The brain sections were washed and incubated with 0.01% Triton X-100 containing 0.5% BSA in 100 mM PBS buffer for 30 min at room temperature. The sections were then incubated with primary antibodies at 4 °C for 2 days. The sections were subsequently incubated with biotin or fluorescence-conjugated secondary antibodies for 1 h at room temperature. The sections were then incubated with the avidin–biotin reagent (VECTASTAIN^®^; ABC kit, Vector Laboratories, Burlingame, CA, USA) and visualized using 0.5 mg/mL DAB (Sigma-Aldrich) in 0.1 M PB containing 0.003% H_2_O_2_. The sections for immunofluorescence were incubated with fluorescent secondary antibodies and then analyzed under a fluorescence microscope (Carl Zeiss, Jena, Germany) and spectral confocal microscope imaging system (A1 Rsi/Ti-E, Nikon).

### 2.7. Western Blot

The expression levels of the target proteins in the DG were quantified through Western blot analyses, as described previously. Briefly, the DG region from the right hemisphere was cautiously dissected using a brain matrix (Roboz Surgical Instrument Co., Gaithersburg, MD, USA). The tissue lysates were centrifuged at 14,000 × *g* for 20 min at 4 °C to remove debris. The supernatants were transferred to a new tube, and the protein concentrations were determined by bicinchoninic acid assay (Bio-Rad Laboratories, Hercules, CA, USA). Then, 50 µg of proteins per each lane were separated on 8% or 15% sodium dodecyl sulfate-polyacrylamide gel (Bio-Rad Laboratories, Hercules, CA, USA) and transferred to 0.45 μm pore size polyvinylidene difluoride membranes (Millipore, Burlington, USA). The blots were probed overnight at 4 °C with the following primary antibodies: rabbit anti-AEG-1 (1:1000, Invitrogen), rabbit anti-GFP (1:500, Millipore), rabbit anti-HA (1:1000, Cell Signaling), rabbit anti-Microtubule-associated protein 1A/1B-light chain 3B (LC3B, 1:1000, Cell Signaling), rabbit anti-p62/SQSTM1 (1:1000, Sigma), rabbit phospho-AMP-activated protein kinase alpha (p-AMPKα, 1:1000, Cell Signaling), mouse anti-AMPKα (1:1000, Cell Signaling), rabbit anti-p70S6K (1:1000, Cell Signaling), rabbit anti-p-p70S6K (1:1000, Cell Signaling), rabbit anti-4E-BP1 (1:1000, Cell Signaling), and rabbit anti-p-4E-BP1 (1:1000, Cell Signaling). After secondary antibody incubation for 1 h at room temperature, the membranes were developed with Pierce ECL Plus (Thermo Fisher Scientific). For quantitative analyses, the density of each band was measured using a computer imaging device and accompanying software (Fuji Film, Tokyo, Japan) and normalized as described.

### 2.8. Statistics

Statistical analyses were performed using one-way ANOVA or two-way ANOVA. The tests were performed depending on comparison variables with Tukey’s post hoc analysis as indicated (Prism; GraphPad Software version 8.3.0, San Diego, CA, USA). All of the values presented are means ± SEM.

## 3. Results

### 3.1. AEG-1 Upregulation in Granule Cells of the DG after KA Treatment

The hippocampal sclerosis of TLE is frequently associated with drug resistance accompanied by neuronal loss in the CA1, CA3, and hilar regions of the DG, as well as GCD [5,30]. Thus, we examined whether KA treatment induces hippocampal sclerosis via NeuN immunostaining one week after KA treatment (Figure 1A). In line with the previous reports [7], we observed that the KA-induced mouse model of TLE recapitulates the characteristic hippocampal sclerosis of TLE evidenced by the loss of neurons in the CA1, CA3, and hilar regions of the DG (Figure 1B). Next, to investigate the possibility of whether there is a regulatory interaction between AEG-1 expression and the epileptogenesis of TLE, we measured the change in AEG-1 expression in the hippocampus of the KA-induced mouse model of TLE. AEG-1 expression is downregulated in the CA1, CA3, and hilar regions of the DG after KA treatment (Figure 1C). Interestingly, its expression is significantly increased in granule cells of the DG in the KA-treated mice compared to the controls (Figure 1C,D). Accordingly, only DG regions were dissected and analyzed by Western blots for the quantitative analysis of AEG-1 protein levels. As demonstrated by Western blotting, the levels of AEG-1 are increased in the DG in a time-dependent manner following KA treatment (Figure 1E), raising a possibility that AEG-1 is involved in the pathogenesis of TLE induced by KA.

### 3.2. Effects of Modulating AEG-1 on Seizure Onset in a KA-Induced Mouse Model of TLE

To examine whether modulating AEG-1 expression in the DG regulates susceptibility to seizure, we measured the latency to seizure onset upon KA treatment at three weeks after an intracranial injection of AAV1 carrying HA-tagged *AEG-1* (AAV1-*AEG-1*) into the DG (Figure 2A). After the AAV1 injection, the mice were sacrificed three weeks later, and the brain tissues were processed to check the levels of AEG-1 in the DGs by using Western blot analyses. We confirmed a significant increase in AEG-1 levels in the DG of the mice infected with AAV1-*AEG-1* compared to the mice infected with AAV1-GFP according to Western blot analyses (Figure 2B). In addition, we observed that the induction of AEG-1 in the DG significantly delays the onset of seizure induced by KA treatment (Figure 2C). Moreover, the induction of AEG-1 in the DG significantly decreases the severity and duration of seizures (Figure 2C). In contrast, the silencing of *AEG-1* by an intracranial injection of AAV1 carrying *AEG-1*-specific shRNA (AAV1-sh*AEG-1*) into the DG markedly decreases AEG-1 levels in the DG (Figure 2D) and the latent time to seizure onset (Figure 2E) and increases the severity of seizures induced by KA treatment (Figure 2E). Thus, our results demonstrated that AEG-1 significantly attenuates the onset and severity of seizure in a KA-induced mouse model of TLE.

### 3.3. Inhibition of mTORC1 Activation by AEG-1 in the DG upon KA Treatment

We next sought to identify the mechanism to link AEG-1 to the attenuation of seizure onset in a KA-induced mouse model of TLE. Numerous studies have demonstrated that the hyperactivation of the mammalian target of rapamycin complex 1 (mTORC1) is a key molecular mechanism underlying the epileptogenesis of TLE [4,31,32,33]. In addition, pharmacological and genetic inhibitions of mTORC1 have been reported to suppress the epileptogenesis of TLE in various rodent models of TLE [27,34,35]. Therefore, we focused on whether AEG-1 regulates mTORC1 activation upon KA treatment in the DG. To exclude the effects of endogenous AEG-1 upregulation triggered by the KA treatment, we conducted Western blot analyses with DG tissues harvested on day 1 after the KA injection. In line with the previous reports, we observed increases in the levels of p-4E-BP1 and p-p70S6K expression, well-established markers of mTORC1 activation, in granule cells of the DG after KA treatment (Figure 3A–C). Of note, we observed that AEG-1 overexpression markedly represses the induction of both p-4E-BP1 and p-p70S6K by KA in the DG compared to control, suggesting that AEG-1 is a negative regulator of mTORC1 (Figure 3C,D). We next investigated whether AEG-1 modulates upstream pathways that regulate mTORC1 activation, such as the PI3K/AKT-, ERK-, and AMPK pathways. Without any effect on the activation of AKT and ERK, we observed that AEG-1 overexpression activates AMPK, evidenced by the induction of p-AMPK, which is inhibited by KA in the DG (Figure 3E), supporting that AEG-1 may suppress mTORC1 activity via activating AMPK pathway. We further assessed if AEG-1 modulates autophagy, which is regulated by mTORC1 under basal conditions. However, we observed that AEG-1 does not regulate the levels of p62 and LC3 lipidation in the DG under KA-treated conditions (Figure 3F).

### 3.4. Attenuation of GCD and SRS by AEG-1 Induction in a KA-Induced Mouse Model of TLE

Rapamycin, a potent selective inhibitor of mTORC1 [25,26,27], is reported to suppress GCD and seizures in rodent models of TLE induced by pilocarpine or KA administration [34,35]. In line with the previous reports, we also confirmed that rapamycin administration one day prior to KA treatment significantly suppresses mTORC1 activation in the DG (Figure 4A,B). Moreover, rapamycin administration significantly suppresses GCD in the DG by KA treatment compared to controls (Figure 4C). Taken together, we confirmed that mTORC1 hyperactivation is a key mechanism underlying GCD in a KA-induced mouse model of TLE. In accord with rapamycin, the overexpression of AEG-1 significantly suppresses GCD in the DG induced by KA treatment (Figure 5A,B). To see if the overexpression of AEG-1 has a long-term therapeutic potential, we next measured recurrent seizure frequency for two weeks from three weeks after KA treatment (Figure 5A,C). Importantly, we demonstrated that the overexpression of AEG-1 significantly attenuates SRS in a KA-induced mouse model of TLE compared to controls, evidenced by a reduction in the frequency, severity, and duration of the seizure (Figure 5C). Taken together, our results suggest that upregulation of AEG-1 in the DG suppresses GCD and SRS by inhibiting mTORC1 hyperactivation in a KA-induced mouse model of TLE.

## 4. Discussion

Epilepsy is induced by various factors that may include infectious, genetic, metabolic, immunologic, and unknown etiologies [4]. Although the causes of epilepsy are quite heterogenous, an imbalance between excitatory and inhibitory neuronal activity has been suggested as a common ictogenic mechanism for epilepsy [36,37]. Based on this theory, more than 20 different ASDs have been used in the clinic that focus on inhibiting neuronal excitability [3,37]. However, almost 30% of epilepsy patients respond poorly to ASD treatments [38]. Moreover, in many cases, patients experienced various adverse side effects, such as dizziness, gastrointestinal problems, depression, cognitive impairments, and psychotic episodes [37].

To address the unmet clinical needs for an epilepsy cure, it is critical to understand the molecular mechanisms underlying epileptogenesis. Several studies have reported mutations in LGI1 and Reelin to be associated with autosomal-dominant epilepsy [39,40,41,42]. As noted, proteolytic processing of Reelin is critical for the maintenance of granule cell lamination in the DG and the regulation of the GCD in the model of TLE [43,44,45,46,47,48,49], a chronic form of focal epilepsy that is often associated with SRS and characteristic hippocampal sclerosis [4,37,50,51]. As mentioned previously, hippocampal sclerosis in TLE is frequently associated with drug resistance and involves neuronal loss in the CA1, CA3, and hilar regions and GCD in the DG [5]. Although the presence and severity of GCD are variable among TLE patients, GCD is known to be closely associated with onset and duration of epilepsy [5,52,53] and is often characterized by abnormal changes in the cytoarchitecture in the GCL, including enlargement of the GCL, a loss of close apposition between granule cells, and the presence of dispersed granule cells in the molecular layer of the DG [5,38]. However, the role of GCD in epilepsy has been challenged in a recent study conducted by Roy and colleagues [54]. Their observation in 147 cadaveric pediatric human samples suggests that the GCD is within the spectrum of normal variation and not unique to patients with epilepsy [54], even though the study cohort is limited to the autopsy series of largely pediatric patients, which may impact their conclusion [54].

In the case of the KA-induced mouse model of TLE, it has been well validated that the GCD is well-correlated with epileptic phenotypes. Of course, although this KA-induced TLE mouse model does not fully reflect the complexity of TLE observed in clinical practice, we believe it has sufficient utility as a model that provides basic biological clues related to GCD of TLE. In any case, accumulating evidence has demonstrated that mTORC1 hyperactivation is a central mechanism to mediate GCD in a KA-induced model of TLE, leading to the development of SRS [4,51,55]. Although rapamycin does not display any effects on the epileptic activity measured with the intrahippocampal EEG in the KA model [55], it is reported to suppress the frequency of seizure behavior in KA-induced rodent models of TLE [34,35,55]. In line with the previous findings, we validated that KA induces mTORC1 hyperactivation in the DG and that inhibition of mTORC1 hyperactivation by rapamycin suppresses GCD induced by KA in vivo. We also demonstrated that AEG-1 upregulation by AAV1-*AEG-1* delivery in the DG significantly suppresses mTORC1 activation. We further demonstrated that induction of *AEG-1* also attenuates GCD, seizure onset, and SRS induced by KA treatment. It is worth noting that the inhibition of AEG-1 in DG increased the severity of seizures induced by KA treatment. Taken together, our findings suggest that AEG-1 significantly attenuates epileptogenesis by suppressing mTORC1 activation in a KA-induced mouse model of TLE. These results suggest one possibility to explain how *AEG-1* can control epileptic seizures and GCD in KA-induced models precisely. Although we demonstrated that *AEG-1* attenuates phenotypic and behavioral abnormality in a KA-induced mouse model of TLE, electrophysiological characterization will be necessary to see if *AEG-1* modulates neuronal excitability. Given that abnormal neuronal excitability is a common ictogenic mechanism for epilepsy, one could envisage a future investigation to see the regulatory role of AEG-1 in neuronal excitability and structural alterations of neural cells at the cellular level. Moreover, the findings of this study revealing that the development of granule cell dispersion happens later and up to three weeks after KA could be adapted to investigate several other underlying pathways and molecules that may be influenced, and their regulation may lead to develop therapeutic strategies against epilepsy.

AEG-1 is ubiquitously expressed across the body [56]. According to the transcriptomic analyses with different neural cells from human and mouse brains, AEG-1 is expressed in all neural cell types [57,58]. Although AEG-1 was initially identified in human fetal astrocytes [15], it has been heavily studied in multiple tumors ever since its upregulation was reported in mouse breast cancer cells [59]. It is now well established that AEG-1 functions as a potent oncogene in multiple tumors [8,9]. AEG-1 is known to activate the PI3K/AKT pathway via a mechanism not yet identified in tumor cells, as well as in astrocyte cells [9,60,61]. In addition, silencing *AEG-1* induces apoptotic cell death by inhibiting PI3K/AKT signaling in mouse motor neuronal NSC34 cells [20]. Previously, we demonstrated that AEG-1 upregulation protects dopaminergic neurons against 6-OHDA-induced neurotoxicity in vivo [19]. Nevertheless, the induction of AEG-1 upregulation does not activate AKT, suggesting that the protective effect of AEG-1 against 6-OHDA in dopaminergic neurons is not dependent on PI3K/AKT signaling [19]. Moreover, in this study, we observed that the inhibition of mTORC1 via AEG-1 upregulation was independent of AKT or ERK signaling in granule cells in the GCL of DG caused by KA administration. These observations strongly argue that the regulatory roles of AEG-1 in the PI3K/AKT pathway are dependent on the cell type and pathological contexts.

Our expanded investigation revealed that exogenously administered AEG-1 results in phosphorylation of p-AMPK. Consequently, this activation significantly inhibited the downstream p-4E-BP1 and S6K proteins. We also confirmed our findings by employing rapamycin, a potent mTOR inhibitor. Taken together, the observed decrease in mTOR and S6 phosphorylation implicates a potential mechanism by which AEG-1 influences cellular processes by regulating AMPK signaling in a KA model of TLE. Autophagy is well-known to be regulated by mTORC1. Of note, a previous study has revealed AEG-1 can promote protective autophagy through noncanonical pathways and activate AMPK [62]. Interestingly, our data showed that AEG-1 does not regulate p62 and LC3 lipidation in the DG under KA-treated conditions, raising the possibility that AEG-1 does not affect autophagy in the KA-induced model of TLE. However, we cannot draw a solid conclusion about whether AEG-1 regulates autophagy in this animal model since the changes in p62 and LC3 are quite variable depending on the context of autophagy dysregulation. That said, further investigations are warranted to elucidate the intricate molecular mechanisms underlying these interactions and to explore the potential therapeutic implications of AEG1.

Taken together, our objective in this study was to investigate the role of AEG1 in the DG of the KA-induced mouse model of TLE. We have confirmed the functional importance of AEG1 in the epileptogenesis of TLE induced by KA in vivo. In the process, we demonstrated that while the inhibition of AEG-1 expression increases susceptibility and severity to seizures, the upregulation of AEG-1 in the DG attenuates GCD and seizure development in a KA-induced mouse model of TLE by inhibiting mTORC1 hyperactivation via AMPK. Though the detailed mechanisms underlying AMPK regulation by AEG1 still need to be clarified, our results suggest that AEG-1 may represent a novel therapeutic target for TLE.

## Figures and Tables

**Figure 1 biomolecules-14-00380-f001:**
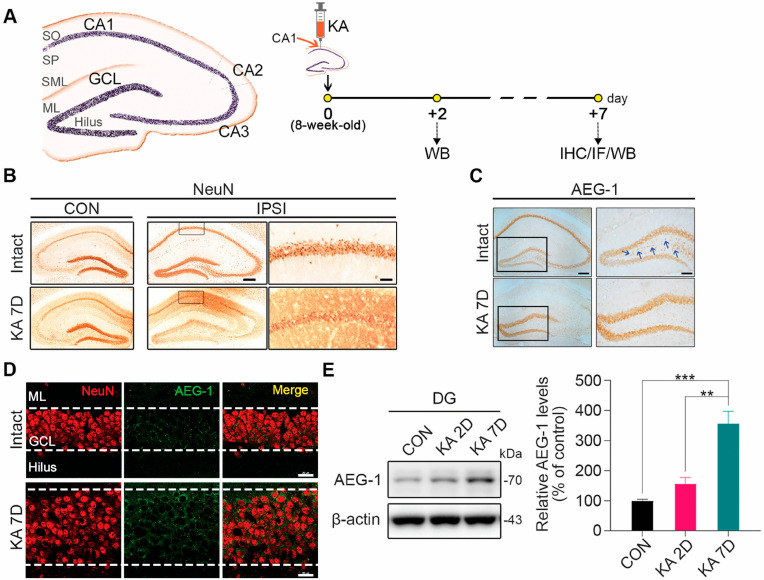
AEG-1 is upregulated in granule cells of DG upon KA treatment. (**A**) Experimental scheme. Each mouse was unilaterally administrated with KA in the right hippocampal CA1 layer by stereotaxic injection. CA, cornu ammonis; SO, stratum oriens; SP, stratum pyramidale; SML, stratum lacunosum-moleculare; ML, molecular layer of the dentate gyrus; GCL, granule cell layer. (**B**) Representative images of NeuN immunostaining of hippocampus at ipsilateral (IPSI) and contralateral sites (CON) of KA treatment. KA treatment induces hippocampal sclerosis at ipsilateral site (scale bars, 200 μm; *n* = 4 in each group). The enlarged images of the rectangular boxes at ipsilateral sites show loss of neurons in the CA1 layer upon KA treatment (scale bar, 50 μm). (**C**) AEG-1 levels are increased in the DG region of hippocampus at ipsilateral site upon KA treatment. AEG-1 immunostaining was performed 7 days after KA treatment (scale bar, 200 μm; *n* = 4 in each group). The enlarged images of the rectangular boxes show GCD and loss of AEG-1 positive signals in hilar neurons (blue arrow) (scale bar, 100 μm). (**D**) KA-induced AEG-1 upregulation is enriched in GCL of the DG. Double immunostaining with NeuN (red) and AEG-1 (green) shows that AEG-1 is upregulated specifically in neurons of GCL in the DG (scale bar, 100 μm; *n* = 4 in each group). (**E**) AEG-1 levels in the DG are increased in a time-dependent manner following KA treatment. Each AEG-1 level was normalized to the corresponding β-actin level and quantified as a percentage of control (*n* = 4 in each group, one-way ANOVA; F(2,9) = 24.24, *p* = 0.0002; CON vs. KA 7D, *p* = 0.0003; KA 2D vs. KA 7D, *p* = 0.0015). Values are mean ± SEM (** *p* < 0.01, *** *p* < 0.001). Original images can be found in Appendix A.

**Figure 2 biomolecules-14-00380-f002:**
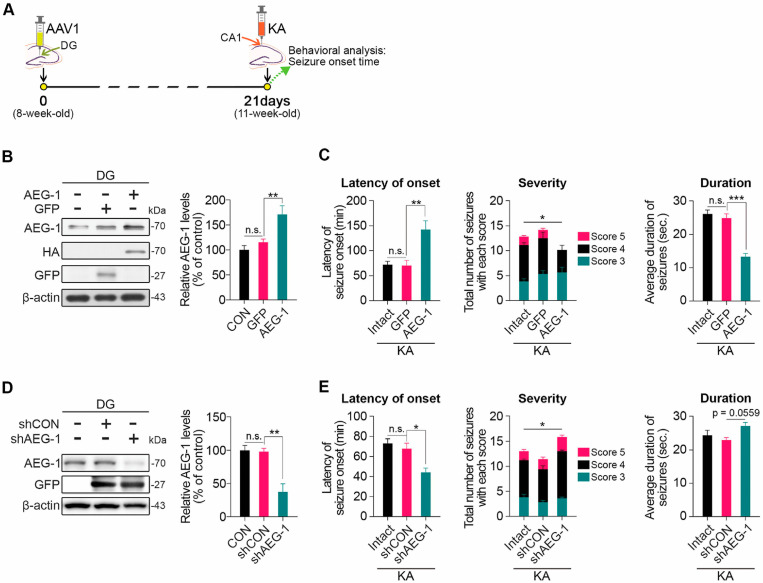
AEG-1 attenuates seizure onset in a KA-induced mouse model of TLE. (**A**) Experimental scheme for behavioral assessment. KA was administered to the right CA1 three weeks after AAV1 injection into the right DG and further seizure onset was assessed up to 4 h following KA treatment. (**B**) *AEG-1* overexpression significantly increased AEG-1 protein levels in the DG of mice injected with AAV1-*AEG-1* (*n* = 4, one-way ANOVA; F(2,9) = 9.954, *p* = 0.0052; GFP vs. AEG-1, *p* = 0.0219). Each AEG-1 level was normalized to the corresponding β-actin level. Original images can be found in Appendix A. (**C**) AEG-1 overexpression attenuates the onset and severity of seizure. Latency of seizure was measured by the latent time to develop the first seizure with severity of score 3 or higher (*n* = 6 in each group, one-way ANOVA; F(2,15) = 11.19, *p* = 0.0011; GFP vs. AEG-, *p* = 0.0024). The severity of each seizure developed up to 2 h from the first seizure was scored based on the scoring system (*n* = 6 in each group, two-way ANOVA interaction effect; F(4,45) = 3.051, *p* = 0.0262; Score 4 GFP vs. AEG-1, *p* = 0.0273). The duration of seizure of each mouse is an average duration time of all seizures developed up to 2 h from the first seizure (*n* = 6 in each group, one-way ANOVA; F(2,15) = 36.10, *p* < 0.0001; GFP vs. AEG-1, *p* < 0.0001). (**D**) Silencing *AEG-1* significantly decreased AEG-1 protein levels in the DG of mice injected with AAV1-shAEG-1 compared to control (*n* = 4 in each group, one-way ANOVA; F(2,9) = 16.69, *p* = 0.0009; shCON vs. shAEG-1, *p* = 0.0021). Each AEG-1 level was normalized to the corresponding β-actin level. Original images can be found in Appendix A. (**E**) *AEG-1* silencing significantly decreases the latency of seizure onset (*n* = 5, one-way ANOVA; F(2,12) = 10.49, *p* = 0.0023; shCON vs. shAEG-1, *p* = 0.0108) and enhances severity of seizure compared to control (*n* = 5, two-way ANOVA interaction effect; F(4,36) = 2.775, *p* = 0.0415; Score 4 shCON vs. shAEG-1, *p* = 0.0001). *AEG-1* silencing also showed a trend to increase the duration of seizure compared to control (*n* = 5, one-way ANOVA; F(2,12) = 3.536, *p* = 0.0621; shCON vs. shAEG-1, *p* = 0.0559). All values are represented as mean ± SEM (n.s. = non-significant, * *p* < 0.05, ** *p* < 0.01, *** *p* < 0.001).

**Figure 3 biomolecules-14-00380-f003:**
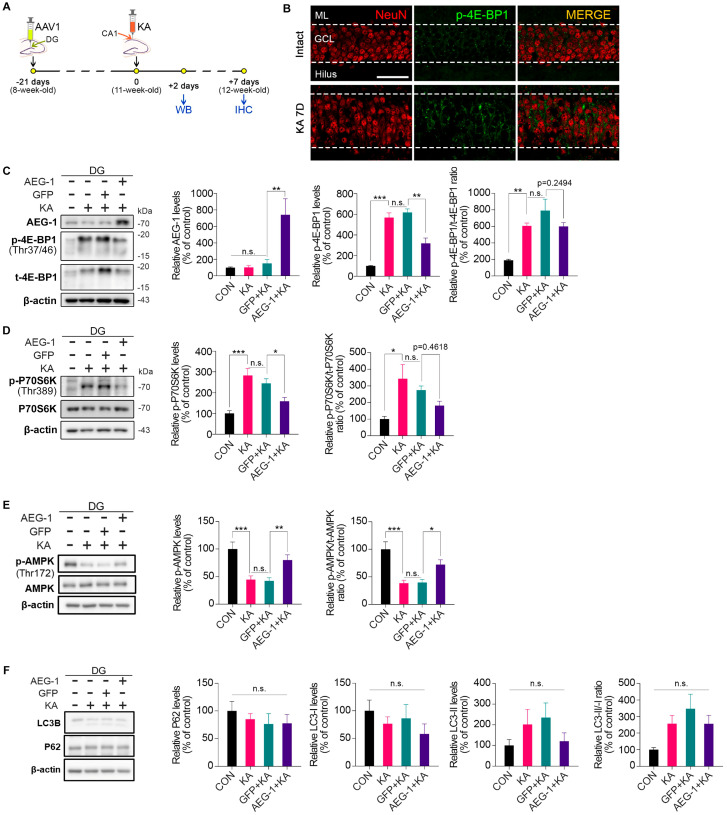
Overexpression of AEG-1 suppresses mTORC1 activation upon KA treatment. (**A**) AAV1 viruses were injected into the GCL of the right DG 3 weeks before KA injection. Two days after KA treatment, DG regions were obtained for Western blot analyses, and at 7 days after treatment, the ipsilateral hemisphere was processed for immunohistochemical analysis. (**B**) Double immunostaining with NeuN (red) and p-4E-BP1 (green) shows that induction of p-4E-BP1 upon KA treatment is highly enriched in GCL of the DG (Scale bar, 50 μm; *n* = 4 in each group). (**C**) AEG-1 overexpression suppresses induction of p-4E-BP1 by KA treatment in the DG of mouse brains. AEG-1 levels (*n* = 4 in each group, two-way ANOVA; F(3,9) = 12.63, *p* = 0.0014; GFP + KA vs. AEG-1 + KA; *p* = 0.0047); *p*-4E-BP levels (*n* = 4 in each group, two-way ANOVA; F(3,9) = 33.71, *p* < 0.0001; CON vs. KA; *p* = 0.0001; GFP + KA vs. AEG-1 + KA, *p* = 0.0029); p-4E-BP/4E-BP ratio (*n* = 4 in each group, two-way ANOVA; F(3,9) = 14.41, *p* = 0.0009; CON vs. KA, *p* = 0.0075; GFP + KA vs. AEG-1 + KA, *p* = 0.2494) Original images can be found in Appendix A. (**D**) AEG-1 overexpression suppresses induction of p-p70S6K by KA treatment in the DG of mouse brains. p-p70S6K levels (*n* = 4 in each group, two-way ANOVA; F(3,9) = 24.53, *p* = 0.0001; CON vs. KA, *p* = 0.0001; GFP + KA vs. AEG-1 + KA, *p* = 0.0233); p-p70S6K/p70S6K ratio (*n* = 4 in each group, two-way ANOVA; F(3,9) = 6.085, *p* = 0.0151; CON vs. KA, *p* = 0.0138; GFP + KA vs. AEG-1 + KA, *p* = 0.4618) Original images can be found in Appendix A. (**E**) AEG-1 overexpression attenuates inhibition of p-AMPK by KA treatment in the DG of mouse brains. p-AMPK levels (*n* = 4 in each group, two-way ANOVA; F(3,9) = 24.08, *p* = 0.0001; CON vs. KA, *p* = 0.0004; GFP + KA vs. AEG-1 + KA, *p* = 0.0053); p-AMPK/AMPK ratio (*n* = 4 in each group, two-way ANOVA; F(3,9) = 24.42, *p* = 0.0001; CON vs. KA, *p* = 0.0002; GFP + KA vs. AEG-1 + KA, *p* = 0.0162) Original images can be found in Appendix A. (**F**) AEG-1 overexpression does not affect the levels of p62 and LC3 in the DG of mouse brains (*n* = 4 in each group, two-way ANOVA). Each protein level was normalized to the corresponding β-actin level. All values are mean ± SEM and represented as a percentage of intact control (n.s. = non-significant, * *p* < 0.05, ** *p* < 0.01, *** *p* < 0.001) Original images can be found in Appendix A.

**Figure 4 biomolecules-14-00380-f004:**
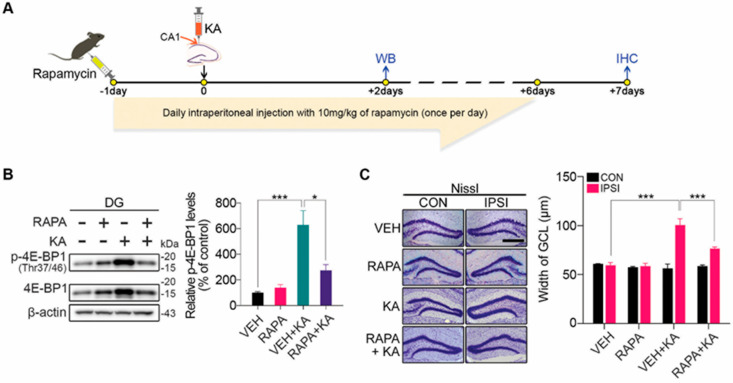
Rapamycin suppresses GCD by inhibiting the hyperactivation of mTORC1 in the GCL of the DG. (**A**) Rapamycin was administered as described in the Section 2 “Materials and Methods”. Two days after KA treatment, DG regions were obtained for Western blot analysis, and at 7 days after treatment, ipsilateral hemispheres were processed for immunohistochemical analyses to assess GCD. (**B**) Rapamycin represses the induction of p-4E-BP1 in the DG upon KA treatment. Each p-4E-BP1 level was normalized to the corresponding β-actin level and quantified as a percentage of intact control (*n* = 4 in each group, two-way ANOVA; F(3,9) = 15.98, *p* = 0.0006; VEH vs. VEH + KA, *p* = 0.0007; VEH + KA vs. RAPA + KA, *p* = 0.0108). (**C**) Rapamycin suppresses GCD induced by KA treatment. The representative images for Nissel-stained sections (scale bar, 500 μm) show that rapamycin administration suppresses GCD induced by KA at ipsilateral site. GCD was assessed by measuring the GCL width of both ipsilateral and contralateral sides 7 days post-KA treatment (*n* = 5 in each group, two-way ANOVA interaction effect; F(3,32) = 21.50, *p* < 0.0001; IPSI VEH vs. VEH + KA, *p* < 0.0001; VEH + KA vs. RAPA + KA, *p* < 0.0001). All values are mean ± SEM (* *p* < 0.05, *** *p* < 0.001) Original images can be found in Appendix A.

**Figure 5 biomolecules-14-00380-f005:**
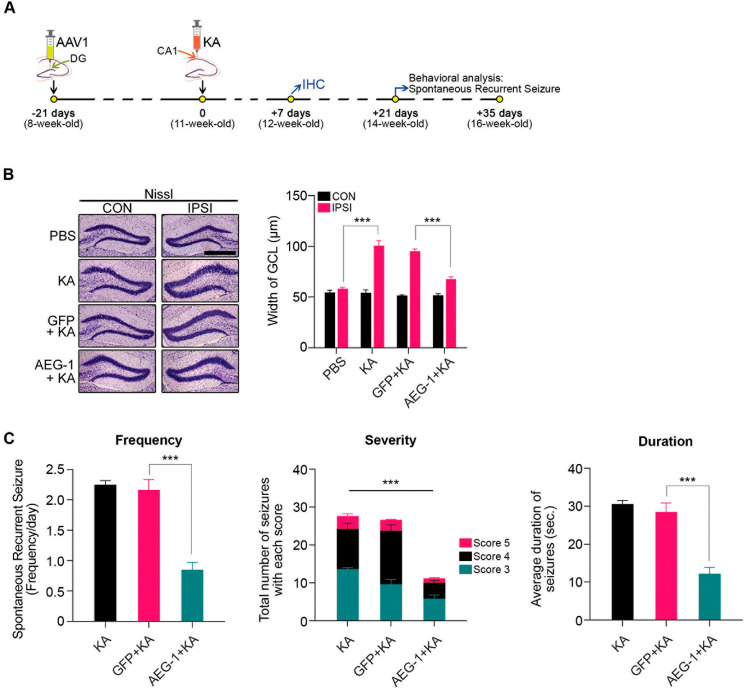
Overexpression of AEG-1 suppresses GCD and SRS. (**A**) AAV1 viruses were injected into the GCL in the right hemisphere. Three weeks later, KA was injected into the CA1 layer of the hippocampus in the right hemisphere. The recurrent seizure frequency was monitored for 2 weeks from 3 weeks after KA treatment. (**B**) AEG-1 overexpression suppresses GCD induced by KA treatment (scale bar, 500 μm). GCD was assessed by measuring the GCL width of both ipsilateral and contralateral sides 7 days post-KA treatment (*n* = 4 in each group, two-way ANOVA interaction effect; F(3,24) = 32.29, *p* < 0.0001; IPSI PBS vs. KA, *p* < 0.0001; GFP + KA vs. AEG-1 + KA, *p* < 0.0001). (**C**) AEG-1 overexpression attenuates SRS induced by KA treatment. From three weeks after KA treatment, the frequency of SRS was measured for two weeks (9 h/day, 6 days/week) using video recording (*n* = 5 in each group, two-way ANOVA; F(2,12) = 37.32, *p* < 0.0001; GFP + KA vs. AEG-1 + KA, *p* < 0.0001). The severity of each seizure was scored based on the scoring system as described in the Section 2 “Materials and Methods” (*n* = 5 in each group, two-way ANOVA interaction effect; F(4,36) = 7.547, *p* = 0.0002; Score 3 GFP + KA vs. AEG-1 + KA, *p* = 0.0222; Score 4 GFP + KA vs. AEG-1 + KA, *p* < 0.0001). The duration of seizure of each mouse is the average duration time of all seizures developed within the monitoring period (*n* = 5 in each group, one-way ANOVA; F(2,12) = 32.29, *p* < 0.0001; GFP + KA vs. AEG-1 + KA, *p* < 0.0001). All values are mean ± SEM (n.s. = non-significant, *** *p* < 0.001).

## Data Availability

All the data are available from the corresponding authors upon reasonable request.

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
