# Peer review of "Inhibition of Granule Cell Dispersion and Seizure Development by Astrocyte Elevated Gene-1 in a Mouse Model of Temporal Lobe Epilepsy"

_biomolecules, 2024, doi:10.3390/biom14030380_

Round 1

Reviewer 1 Report

Comments and Suggestions for Authors

The data presented are new and very interesting. The fact is established (with respective experimental designs) that the AEG-1 gene expression is connected with the reduction of seizures in kainic acid murine model. The discussion section containes the interesting informatuion is well which makes the data  presented more convincing.The paper is almost ready for publication although some minor critical  notes could be given. For instance, TLE as human disease is much more complecated (including  mental symptoms as well) that the murine model, used in this study. This  moment is worth to be mentioned in the text.   The figure legends are two long and part of respective information could be transferred into the text. The experimental data were obtained in well planned experimental series and should be published as they represent interest to  those, who are working  in the field of new approaches to epilepsy therapy and to those scientists, who are investigationg the bases of epileptogenesis in general. The paper is worth to be published.

Comments on the Quality of English Language

No special comments

Author Response

Reviewer #1: The data presented are new and very interesting. The fact is established (with respective experimental designs) that the AEG-1 gene expression is connected with the reduction of seizures in kainic acid murine model. The discussion section containes the interesting informatuion is well which makes the data  presented more convincing.The paper is almost ready for publication although some minor critical notes could be given. For instance, TLE as human disease is much more complecated (including  mental symptoms as well) that the murine model, used in this study. This  moment is worth to be mentioned in the text.  The figure legends are two long and part of respective information could be transferred into the text. The experimental data were obtained in well planned experimental series and should be published as they represent interest to  those, who are working  in the field of new approaches to epilepsy therapy and to those scientists, who are investigationg the bases of epileptogenesis in general. The paper is worth to be published.

Response: We would like to thank the reviewer for their thoughtful review and the comments, corrections, and suggestions of the manuscript. A revision of the paper has been carried out concerning the advised corrections to take all of them into account. Point-by-point responses to all comments of the reviewer are given below and are incorporated into the revised manuscript (red font).

  1. Comment: For instance, TLE as human disease is much more complecated (including  mental symptoms as well) that the murine model, used in this study. This  moment is worth to be mentioned in the text.  

Response: We appreciate the reviewer’s comment and agree to the fact that TLE in humans is more complex than the reproduced TLE in animal models. As suggested, we have now discussed this fact on page 12; lines 404 - 409).

  1. Comment: The figure legends are two long and part of respective information could be transferred into the text.

Response: We appreciate the reviewer's feedback. As suggested, we have attempted to shorten the figure legends and integrate the relevant information within the text.

Reviewer 2 Report

Comments and Suggestions for Authors

The subject of the manuscript by Eunju et al. is the impact of astrocyte elevated gene-1 (AEG-1) to the pathophysiology of TLE. Interestingly this well-known oncogene is not only implicated in various types of cancers, but also is getting increasingly relevant to fundamental biological functions and pathophysiological processes. By using histological, immunohistochemical, molecular biological and behavioural test the authors implicate AEG-1 for the first time in epilepsy bringing up the modulation of AEG-1 as a potential ASD treatment.

The manuscript is well written and data as well as methods are sound. Although reading the manuscript attentively I could only find some minor concerns that need to be addressed.

 (1) The authors used several abbreviations in the title. To reach a wide range of readers I would suggest simplifying the title. For example: “Inhibition of granule cell dispersion and seizure development by astrocyte elevated gene-1 (AEG-1) in a mouse model of temporal lobe epilepsy”

(2) By reading the introduction I was missing some information on expression and function of AEG-1 under physiological condition.

(3) I appreciate the presented figures that are rich in content. Nevertheless, some graphs (especially the immunohistochemical and histological figures) are too small and/or the resolution is too low in my point of view. I would kindly ask the authors to check the figures and improve the readability if possible.

(4) What does the sample size indicate (slices, individuals etc.)?

(5) I could not find any statistical analysis by t-test even thought it is mentioned in the “materials and methods” section.

(6) With reference to Section “Effects of modulating AEG-1 on seizure onset in a KA-induced mouse model of TLE” (page 6) and figure 2 (page 7)

Please mention at which particular time these western blot experiments have been conducted. Otherwise, it’s not possible to assign the elevation of AEG-1 to the AVV-1 infection. Maybe the timepoint of WB could be indicated in figure 2A?

Author Response

Reviewer #2: The subject of the manuscript by Eunju et al. is the impact of astrocyte elevated gene-1 (AEG-1) to the pathophysiology of TLE. Interestingly this well-known oncogene is not only implicated in various types of cancers, but also is getting increasingly relevant to fundamental biological functions and pathophysiological processes. By using histological, immunohistochemical, molecular biological and behavioural test the authors implicate AEG-1 for the first time in epilepsy bringing up the modulation of AEG-1 as a potential ASD treatment.

The manuscript is well written and data as well as methods are sound. Although reading the manuscript attentively I could only find some minor concerns that need to be addressed.

Response: We thank the reviewer for their positive feedback. An attempt has been made to incorporate their suggestions into the revised manuscript (red font).

  1. Comment: The authors used several abbreviations in the title. To reach a wide range of readers I would suggest simplifying the title. For example: “Inhibition of granule cell dispersion and seizure development by astrocyte elevated gene-1 (AEG-1) in a mouse model of temporal lobe epilepsy”

Response: We thank the reviewer for their suggestion. As suggested, the title has been modified to “Inhibition of granule cell dispersion and seizure development by astrocyte elevated gene-1 in a mouse model of temporal lobe epilepsy”.

  1. Comment: By reading the introduction I was missing some information on expression and function of AEG-1 under physiological condition.

Response: We appreciate the reviewer’s comment. As suggested, we have added information with respect to the expression and function of AEG-1 under the physiological condition in the introduction (page 2; lines 60 - 65).

  1. Comment: I appreciate the presented figures that are rich in content. Nevertheless, some graphs (especially the immunohistochemical and histological figures) are too small and/or the resolution is too low in my point of view. I would kindly ask the authors to check the figures and improve the readability if possible.

Response: We apologize for the inconvenience. We have revised the resolution (600 dpi) and text in the figures and graphs at appropriate place to enhance the clarity.

  1. Comment: What does the sample size indicate (slices, individuals etc.)?

Response: The sample size indicates the number of individuals. We have also included the sample size for histological data in the respective figure legends.

  1. Comment: I could not find any statistical analysis by t-test even through it is mentioned in the “materials and methods” section.

Response: We apologize for the inconvenience and also thank the reviewer for their thorough review of our manuscript and for pointing out the error. We have now revised the incorrect expression in the section: Materials and Methods (page 4; lines 196 - 199).

  1. Comment: With reference to Section “Effects of modulating AEG-1 on seizure onset in a KA-induced mouse model of TLE” (page 6) and figure 2 (page 7)

Please mention at which particular time these western blot experiments have been conducted. Otherwise, it’s not possible to assign the elevation of AEG-1 to the AVV-1 infection. Maybe the timepoint of WB could be indicated in figure 2A?

Response: We thank the reviewer for their comment and apologize for the inconvenience. As suggested, all figure legends including Figure 2A have been modified with respect to the time for the Western blot experiments.